# Dexamethasone and Dexamethasone Phosphate: Effect on DMPC Membrane Models

**DOI:** 10.3390/pharmaceutics15030844

**Published:** 2023-03-04

**Authors:** Candelaria Ines Cámara, Matías Ariel Crosio, Ana Valeria Juarez, Natalia Wilke

**Affiliations:** 1Departamento de Fisicoquímica, Facultad de Ciencias Químicas, Universidad Nacional de Córdoba, Córdoba X5000HUA, Argentina; 2Instituto de Investigaciones en Fisicoquímica de Córdoba (INFIQC), Consejo Nacional de Investigaciones Científicas y Técnicas (CONICET), Córdoba X5000HUA, Argentina; 3Departamento de Química Biológica Ranwel Caputto, Facultad de Ciencias Químicas, Universidad Nacional de Córdoba, Córdoba X5000HUA, Argentina; 4Centro de Investigaciones en Química Biológica de Córdoba (CIQUIBIC), Consejo Nacional de Investigaciones Científicas y Técnicas (CONICET), Córdoba X5000HUA, Argentina

**Keywords:** Dexamethasone, Dexamethasone phosphate, membrane models, dimyiristoylphophatidylcholine

## Abstract

Dexamethasone (Dex) and Dexamethasone phosphate (Dex-P) are synthetic glucocorticoids with high anti-inflammatory and immunosuppressive actions that gained visibility because they reduce the mortality in critical patients with COVID-19 connected to assisted breathing. They have been widely used for the treatment of several diseases and in patients under chronic treatments, thus, it is important to understand their interaction with membranes, the first barrier when these drugs get into the body. Here, the effect of Dex and Dex-P on dimyiristoylphophatidylcholine (DMPC) membranes were studied using Langmuir films and vesicles. Our results indicate that the presence of Dex in DMPC monolayers makes them more compressible and less reflective, induces the appearance of aggregates, and suppresses the Liquid Expanded/Liquid Condensed (LE/LC) phase transition. The phosphorylated drug, Dex-P, also induces the formation of aggregates in DMPC/Dex-P films, but without disturbing the LE/LC phase transition and reflectivity. Insertion experiments demonstrate that Dex induces larger changes in surface pressure than Dex-P, due to its higher hydrophobic character. Both drugs can penetrate membranes at high lipid packings. Vesicle shape fluctuation analysis shows that Dex-P adsorption on GUVs of DMPC decreases membrane deformability. In conclusion, both drugs can penetrate and alter the mechanical properties of DMPC membranes.

## 1. Introduction

Dexamethasone (Dex) and Dexamethasone phosphate (Dex-P), Figure 1a,b, are synthetic glucocorticoids derived from cortisol, an endogenous glucocorticoid with anti-inflammatory action. These drugs are thirty times more potent than cortisol and have a lower mineralocorticoid effect due to the presence of a flour atom in its structure [1,2]. The active drug in the organism corresponds to Dex, while Dex-P is its pro-drug, being converted to Dex by the action of alkaline phosphatase [3]. After intravenous administration, the phosphate ester of Dex-P hydrolyzes with a half-life time of 5 min, reaching the maximum Dex concentrations in plasma after 10 min [4]. The more important difference between Dex and Dex-P is the higher aqueous solubility of Dex-P.

In recent decades, Dex and Dex-P have been widely used in several diseases due to their anti-inflammatory and immunosuppressive activity [5]. Both drugs are used in patients with solid tumors to inhibit tumor progression, and as adjuvant in chemotherapy, to reduce vomit, nausea, and fatigue [6]; in ocular disease [7,8]. Additionally, they are used for treating several allergy episodes [9] and in local inner ear therapies [10]. The application of these glucocorticoids in occlusive airway conditions such as asthma attacks is interesting [11]; they are used to enhance the lung maturation on fetuses [12] and in patients with affected lungs due to the infection with COVID-19 [13,14,15]. In all patients recovered from COVID-19 virus infection, the drug was administered orally using an assisted ventilation system that directs the drug to the target organ (lung). Dex arrives at the pulmonary alveolar sacs, interacting with the pulmonary surfactant. These surfactants form a monolayer in the alveolar sacs that contains lipids and proteins, being composed of 80% phosphatidylcoline (PC) [16,17]. This film has the function of decreasing the surface tension in the interface during breathing [18].

Treatment with corticoids involves different periods of drug administration. In the case of an acute condition (e.g., allergic attack), the treatment involves high doses during short periods, while in a chronic illness (e.g., Addison’s disease), the drug doses are lower and for long periods [19]. Chronic administration of steroids promotes side effects such as adrenal insufficiency, hyperglycemia, osteoporosis, and the development of diabetes mellitus [20].

The route of administration of Dex and Dex-P depends on the needs of the patient; they can be distributed via topical, oral, intramuscular, intralesional, or intravenous injection. Due to its high-water solubility, Dex-P doses can be of 10 mg/mL or higher, depending on the infection [18]. Therefore, high local concentrations may be present during treatment.

Dex is expected to alter biological membranes due to its hydrophobic characteristic and structural similitude with hormones. Ghosh et al. found that Dex reduces the fluorescence anisotropy of 1,6 diphenyl 1,3,5 hexatriene (DPH) in the bilayers in the gel state of DPPC and shifts the transition temperature to a lower value, suggesting a fluidizing effect on bilayers [21]. Similar results were obtained by Saija et al. using vesicles and monolayers composed of DMPC [22]. Modi and Anderson determined the water/bilayer partition coefficient of Dex and Dex-P in liposomes of different compositions and pHs. They found a higher membrane/water partition coefficient for Dex than Dex-P, being highly dependent on the bilayer chain length. In the case of Dex-P, the membrane/water partition coefficient decreases at high pHs due to the increase in the degree of ionization of Dex-P [23].

The action of similar corticoids on membranes has been reported. Yi E. Wang et al. studied the effects of budesonide and beclomethasone on Langmuir films composed of commercial pulmonary surfactants [24,25]. They found that a 10% molar fraction of budesonide induces the formation of more and smaller condensed domains, and enhances multilayer formation at high surface pressures [25].

In this work, we report the interaction of Dex and Dex-P with DMPC membranes with the aim of studying how the presence of the drugs modifies the membrane properties. We studied the compressional behavior of Langmuir monolayers composed of DMPC or DMPC:drug (Dex or Dex-P) at different molar fractions, films were observed with Brewster Angle Microscopy while compressed. The insertion of Dex or Dex-P in DMPC monolayers was studied at increasing drug concentration in the aqueous subphase. Besides, we inquired on the influence of the drugs on membrane stiffness analyzing shape fluctuations of giant unilamellar vesicles (GUVs).

The liquid-expanded/liquid-condensed phase transition present in pure DMPC films, was no longer present at Dex molar fractions of 70% or higher. Besides, film reflectivity and stiffness decrease in the presence of Dex. In turn, Dex-P effects on DMPC monolayers were less marked. The cut off value for drug insertion was of 45 mN/m or 36 mN/m for Dex and Dex-P, respectively, indicating that both drugs are able to incorporate to monolayers at lipid densities comparable to those in bilayers [26,27]. Regarding membrane bending rigidity, Dex-P adsorption reduced in 28% the amount of DMPC fluctuating GUVs whilst Dex did not modify membrane fluctuations.

## 2. Materials and Methods

### 2.1. Materials

The phospholipid 1,2-dimirystoyl-sn-glycero-3-phosphocholine (DMPC) and the lipophilic fluorescent probe L-α-phosphatidylethanolamine-N-(lissamine rhodamine B sulfanyl) (ammonium salt) (egg-transphosphatidylated, chicken) (Rho-egg PE) were purchased from Avanti Polar Lipids (Alabaster, Al). Dexamethasone (Dex; M.W 392.46 g/mol) and Dexamethasone 21-phosphate disodium salt (Dex-P; 516.40 g/mol) were purchased from Sigma-Aldrich with a degree of purity of ≥98%, the powder was used without further purification. Glucose, saccharose, ethanol, chloroform, and methanol were also purchased from Sigma-Aldrich. The solvents had the highest commercial purity available.

Lipid solutions used for Langmuir monolayer and insertion experiments were prepared in Cl3CH/CH3OH 2:1 *v*/*v* with a final concentration of 1 nmol/μL and 2.5 nmol/μL, respectively. Dex and Dex-P for Langmuir monolayer experiments were prepared in Cl_3_CH/CH_3_OH with a concentration of 5 nmol/μL and 4 nmol/μL, respectively. For insertion and vesicle experiments, Dex-P was dissolved in NaCl solution (Merck Millipore—Emsure; NaCl 10 mM for GUVs and 1 mM for LUVs experiments) to a final concentration of 25 mM, stock solution. Dex stock solution was prepared in ethanol to a final concentration of 10 mM for GUVs and LUVs experiments and 14.5 mM for insertion experiments (solution in ethanol: water 7:5).

The aqueous solutions were prepared with deionized water (with resistivity of 18 MΩ, obtained from a Milli-Q Gradient System, Millipore, Bedford, MA, USA).

### 2.2. Methods

#### 2.2.1. Monolayer at Air/Liquid Interfaces

Compression isotherms and insertion experiments were performed with a commercial Langmuir balance (mini-trough II from KSV Instruments Ltd., Helsinki, Finland), using the Wilhelmy method with a platinum plate. The compression isotherms were performed by compressing the films at 5 mm/min. The aqueous subphase, contained in a Teflon trough (364 mm × 75 mm effective film area), was 150 mM NaCl, pH = 6.00. The injected mixture of DMPC:X (X: Dex or Dex-P) were made by mixing the desired amounts of each component in Cl_3_CH/CH_3_OH to obtain DMPC:X, 1: 0, 9:1, 7:3, 1:1, 3:7 and 2:8 molar fraction. All mean areas in Langmuir compression isotherms were calculated considering all molecules spreaded onto the subphase (drug + lipids), despite in some experiments not all molecules stayed in the interface. An apparent surface compression modulus, *κ_app_* (mN/m), was calculated from the Langmuir isotherm as:(1)$$\kappa_{app}=-{\left(\frac{d\pi }{dA}\right)}_{T}dA$$
where *A* is the trough area and *π* is the surface pressure in mN/m. This is an apparent compressional modulus because, in some experiments, the film is not a molecule thick, as shown later.

Monolayers were observed during compression using Brewster angle microscopy (BAM) with an EP3 Imaging Ellipsometer (Accurion, Goettingen, Germany) equipped with a 20× objective (Nikon, NA 0.35, Tokyo, Japan). The equipment was calibrated with the clean interface, allowing the calculation of the reflected light intensity (Rp) from the gray-levels of the images. Reproducible images were obtained, the size, shape, and number of the domains for different monolayers were similar, as well as the gray-level of the domains and the continuous phase. The average gray-levels (±SD) were calculated from the gray-levels in 6 different regions corresponding to each phase in at least 4 images for each condition.

The analysis and quantification of the images were performed using ImageJ 1.53t free software.

In the insertion experiments, the lipid solution was spread onto the air/water interface up to the desired surface pressure (in the range of 5–40 mN/m), and the surface pressure was registered after injection of the desired amounts of Dex or Dex-P into the subphase. When the surface pressure reached a plateau (less than 20 min), the change in surface pressure was determined.

The temperature was adjusted by air-conditioning the room and by controlling the temperature of the trough by means of a thermocirculator (ARCTIC AC200-A10 Thermo Fisher Scientific Inc., Waltham, MA, USA). The temperature was kept at 15 °C for isotherm experiments and 20 °C for insertion experiments.

#### 2.2.2. Giant Unilamellar Vesicles (GUVs)

GUVs were prepared by the electroformation technique as described by Angelova and Dimitrov [28] using a home-made wave generator and a chamber with stainless steel electrodes [29]. Briefly, 10 μL of a 0.5 mM lipid solution doped with 0.5 mole% of the fluorescent probe Rho-PE (in chloroform- methanol 2:1) were spread on two stainless steel electrodes, and were left under vacuum for at least 1 h to remove all traces of the organic solvent. The lipid films were hydrated by filling a homemade chamber of acrylic containing the electrodes with a 300 mM sucrose and 10 mM NaCl solution. GUVs were formed in a lower salt concentration than the monolayer experiments to prevent the electrode oxidation in the electroformation process. The electrodes were connected to the wave generator, and a sinusoidal tension of 1–2 V amplitude and 10 Hz frequency was applied for 1 h at 25 °C.

The observation chamber was pretreated with a 10% (*w*/*v*) of albumin solution, which prevented the rupture of GUVs on the glass slide. GUVs were observed with confocal microscopy (Confocal Zeiss FV1000) with a 60× objective (immersion in oil). For this, 40 μL of the GUV suspension was transferred to the observation chamber containing 200 μL of a 300 mM glucose and 10 mM NaCl solution in the absence or presence of Dex (4.8 μL of 10 mM stock solution, final concentration 0.2 mM) or Dex-P (10 μL of 25 mM stock solution, 1 mM). Controls for Dex were created by adding the same volume of ethanol without Dex. Osmolarity parity between sucrose and glucose solutions was tested with an automatic micro-osmometer OM-806 (Vogel, Kevelaer, Germany).

Shape fluctuations were determined for GUVs registering movies of 15−20 individual GUV (200 to 500 frames, 460 frames/min) at each condition. For each vesicle, the circularity value (Cir = 4π area/(perimeter)^2^) as a function of time (Appendix A) was determined using the morphological package of the free software ImageJ, the average circularity ($\bar{Cir}$)′ in the time, and its standard deviation SD($\bar{Cir}$) was statistically analyzed. Each vesicle was classified as “fluctuating” or “non-fluctuating” depending on its value of SD($\bar{Cir}$) compared with a reference value designed as upper limit (UL). We defined the treatment that leads to the less fluctuating GUV population as the treatment with the lower value of percentile 95th (95th, i.e., the value bellow which the 95% of the data is found [30]), see Appendix A. The lower value of P95th was taken as the UL value, and GUVs were classified according to this value as fluctuating (SD($\bar{Cir}$) ≥ UL) or non-fluctuating (SD($\bar{Cir}$) < UL), Appendix A.

The percentage of fluctuating GUVs in each condition was calculated as:(2)$$\%\;\mathrm{fluctuating}\;\mathrm{GUVs}=\left(\frac{\mathrm{number}\;\mathrm{of}\;\mathrm{fluctuating}\;\mathrm{GUVs}}{\mathrm{number}\;\mathrm{of}\;\mathrm{total}\;\mathrm{GUVs}}\right)\times 100$$

This procedure was performed in two independent sets of experiments and the averages ± SE are shown [31]. All the statistical analysis was made using the free statistical software InfoStat 2019 version.

#### 2.2.3. Large Unilamellar Vesicles (LUVs), Z-Potential and Hydrodynamic Size Measurements

LUVs were obtained forming a uniform lipid film on the wall of a glass tube by solvent evaporation under a N_2_ flow from a chloroform–methanol lipid solution. Final traces of solvent were removed by incubating the lipids in a high-vacuum chamber for 1 h. Then, the lipids were resuspended in NaCl 1 mM solution, to a final lipid concentration of 0.3 mM. The suspension was incubated in a 50 °C water bath, after that, it was vortexed and subjected to cold incubation, each step holding 30 s. This procedure was repeated nine times. The resulting multilamellar vesicles were extruded 21-times through a 100 nm pore filter (Avanti) at 50 °C. For z potential and dynamic light scattering (DLS) measurements, the 0.3 mM lipid dispersion was diluted to a final concentration of 0.12 mM. Then, an appropriate volume of a stock solution of the Dex was added to 100 µL of this dilution to achieve the desired Dex/lipid molar ratio (D/L), and incubated for 1 h. The z-potential determinations were performed by means of Henry’s equation, subjecting aliquots of 100 µL of liposomes and drug to an electric field and determining electrophoretic mobility with a Z-sizer SZ-100-Z equipment (Horiba, Ltd., Kyoto, Japan). The temperature was adjusted to 25 °C for all these experiments.

## 3. Results

### 3.1. Insertion Experiments

Figure 1 shows the chemical structure of DMPC, Dex, and Dex-P. Dex-P shares the cyclic structure with Dex, but has the 17-C derivatized with a phosphate group, which gives an acid-base character with pKa_1_ = 1.9 and pKa_2_ = 6.4 [18,23] to the molecule. At the working pH (6.00), Dex-P is negatively charged (−1), DMPC is zwitterionic [23], and Dex is neutral. At this pH, we could evaluate, from a physicochemical point of view, the role of the negative charge present in Dex-P on the interaction with DMPC membranes and compare it with the neutral drug.

In order to study the adsorption of Dex and Dex-P on a pre-formed DMPC monolayer, insertion experiments of both drugs were done. These experiments allow us to characterize the first contact of the drug with the lipidic monolayer. For this, the drug was injected into the aqueous phase and the spontaneous adsorption process to a pre-formed monolayer was followed by the change in surface pressure after the drug addition.

Firstly, we studied the adsorption of Dex and Dex-P on DMPC monolayers as a function of the injected drug concentration. In these experiments, the DMPC monolayer was formed at 15 mN/m (the monolayer is in liquid expanded phase) on NaCl 150 mM and a given amount of the drug was added to the aqueous phase. The surface pressure was registered, and the final value (after stabilization, approximately 20 min) was measured for each concentration. Figure 1a shows the change in surface pressure ($∆\pi_{max}=\pi_{f}-\pi_{i}$) as a function of Dex or Dex-P final concentration.

Since both curves showed a hyperbolic-like behavior, the maximal change in surface pressure ($∆\pi_{max}$) generated by the drugs in DMPC monolayers and the $K_{0.5}$ value (drug concentration needed to reach half the maximal increment in surface pressure, $∆\pi_{max}$/2 were obtained adjusting the curves to the following equation [32]:(3)$$∆\pi =\left(∆\pi_{max}C\right)/\left(K_{0.5}+C\right)\;$$

Table 1 summarizes the results for $∆\pi_{max}$ and $K_{0.5}$ for Dex and Dex-P adsorption on DMPC monolayers. The values which indicate that Dex induced larger changes in surface pressure than Dex-P at all analyzed concentrations, and that the concentration required to induce a 50% of the maximal change in surface pressure was one order of magnitude lower for Dex. The obtained results can be explained considering the higher hydrophobic character of Dex compared to Dex-P, conferring the drug a higher affinity for the monolayer, and thus inducing larger changes in surface packing and saturating the monolayer at lower concentrations. It is worth noting, however, that Dex-P is incorporated into DMPC monolayers despite its negative charge.

Knowing the dependence of the drug penetration with its concentrations, we next determined the influence of DMPC film packing on drug penetration. For this, a fixed drug concentration was used (0.2 mM of Dex or 1.2 mM of Dex-P) and the initial surface pressure of the film ($\pi_{i}$) was varied.

Figure 1b shows the change in surface pressure ($∆\pi_{max}=\pi_{f}-\pi_{i}$) as a function of the initial surface pressure (*π_i_*) after Dex or Dex-P addition. The exclusion pressures ($\pi_{cut\;off}$), given in Table 1, were calculated as the extrapolated value for which ∆*π* = 0 (arrows in the figure), and indicates which is the maximal surface pressure at which the drug is included in the film. The results indicate that Dex was able to penetrate DMPC films to surface pressures 9 mN/m higher than Dex-P, which is in agreement with the results shown up to now.

Both drugs can penetrate the monolayer at high surface pressures, which correspond to lipid densities compared to those in bilayers [24,26].

### 3.2. Langmuir Isotherm

Once the spontaneous adsorption of the drugs on a DMPC monolayer was studied, we inquired about the modification of the monolayer properties induced by Dex or Dex-P by compressing films prepared with mixtures of DMPC and the drug. In order to study the effect of the drug in the LE/LC phase transition of DMPC films, we performed the experiments at 15 °C, since at room temperature (20–25 °C), the phase transition occurs close to film collapse [33,34,35].

Figure 2a,b shows the compression isotherms for DMPC and DMPC:X mixtures (where X: Dex or Dex-P). DMPC monolayer is characterized by a liquid expanded (LE)/liquid condensed (LC) phase transition at (33.4 ± 0.2) mN/m and a collapse pressure and area of (49 ± 1) mN/m and (40 ± 1) Å^2^, respectively. The compressibility modulus (*κ_app_*) value of DMPC films in the LE phase is (70 ± 5) mN/m (at *π* = 20 mN/m), whereas in the LC phase, a *κ* = (103 ± 9) mN/m (at *π* = 40 mN/m), Figure 2c,d.

When pure Dex was spread in the air/water interface, no monolayer was formed, even at high concentrations. However, when Dex was mixed with DMPC, a monolayer with properties different to those of pure DMPC monolayers was formed. Figure 2a, shows Langmuir isotherms for a DMPC: Dex monolayer as a function of the area per spreaded molecules for different Dex molar fractions. At a low Dex molar fraction (0.1), the isotherms slightly shifted towards higher areas per spreaded molecules, giving a collapse area of (43 ± 1) Å^2^. In contrast, a subsequent increment in Dex molar fraction inverted this tendency and the curve shifted towards lower areas per spreaded molecules, arriving to a collapse area of (9 ± 1) Å^2^ for 0.8 Dex fraction. Figure 2e summarizes the tendency in area per spreaded molecules with the increment of Dex molar fraction at a surface pressure near to collapse (*π* = 40 mN/m). Considering the crystallographic structure of Dex [1,36] and its chemical structure, which is similar to that of cholesterol, similar mean molecular areas are expected. Therefore, we assume that Dex occupied the same minimal mean molecular area, which is marked as dashed horizontal lines in Figure 2c (37 Å^2^) [37,38].

Taking into account that the mean molecular area of DMPC films at *π* = 40 mN.m^−1^ and 15 °C is 42 Å^2^, a cross sectional area lower than 37 Å^2^ is not possible for a monomolecular film. Therefore, we propose that desorption of the molecules to the subphase or three-dimensional aggregation occurred at Dex molar fractions higher than 0.3. At low molar fractions (0.1), Dex was non-ideally mixed or partially mixed with DMPC [39], since the collapse surface pressure of the film (*π_collapse DMPC:Dex real_* = 46.6 mN/m) was different from that of pure DMPC (*π_collapse DMPC_* = 50 mN/m) and ideal mixture (*π_collapse DMPC:Dex ideal_* = 49 mN/m).

When we evaluate the Langmuir isotherm considering the molecular area per lipid instead of the area per spreaded molecules (Appendix A), we found that increasing Dex proportion shifts the isotherm towards higher areas compared to pure DMPC, suggesting that DMPC mixes with Dex molecules even when the molar fraction of Dex is 0.8.

In Figure 2a, it can be seen that the LE/LC phase transition of DMPC was modified by the presence of Dex, becoming less marked, but without an evident modification on the transition surface pressure. This can be explained considering a decrease in the cooperativity of the phase transition due to the presence of Dex in the monolayer [40]. For a Dex fraction of 0.7 or higher, the LE/LC phase transition is not evident from the compression isotherm.

Figure 2c displays *κ_app_* as a function of the surface pressure for all DMPC: Dex molar fractions. As can be seen in Figure 2f, at high surface pressure (40 mN/m) the presence of Dex at molar fractions higher than 0.3 significantly decreased sthe compressibility of the monolayer, going from (103 ± 9) mN/m in the absence of Dex to (58 ± 3) mN/m for Dex molar fraction of 0.8. For molar fractions of 0.7 and 0.8, the minimum corresponding to LE/LC phase transition was not evident, in concordance with the lack of plateau in the compression isotherms. Contrary to the decrease in monolayer stiffness observed at high surface pressures, no significant changes in *κ_app_* were observed at surface pressures lower than 30 mN/m.

A similar analysis of the Langmuir compression isotherms was done for films composed of DMPC: Dex-P, Figure 2b. Dex-P alone, similar to Dex, had no surface activity. At low Dex-P molar fractions (0.1), the compression isotherms shifted slightly to larger areas per spreaded molecules at surface pressure above 15 mN/m. Similar to Dex, increasing Dex-P molar fraction led to a shift of the compression isotherms to lower areas per spreaded molecules, which can be explained as a consequence of aggregation or desorption process as for Dex. When the compression isotherms were analyzed as functions of the mean molecular areas per lipid, the DMPC monolayer shifts to larger areas even at high fraction of Dex-P indicating a non-saturating DMPC film (see Appendix A).

Contrary to the effect promoted by Dex, the LE/LC phase transition is evident at all Dex-P proportions (Figure 2b). The minimum in *κ_app_* corresponding to the phase transition can be clearly detected even at high molar fractions of Dex-P. This molecule promotes an increase in the compressibility of the LC phase state, opposite to Dex effects, from (103 ± 9) mN/m to (160 ± 15) mN/m at *π* = 40 mN/m (Figure 2d,f).

### 3.3. Brewster Angle Microscopy

Langmuir films were visualized with Brewster angle microscopy (BAM) during compression. Figure 3 shows the BAM images for DMPC monolayers in the absence and presence of increasing molar fraction of Dex and Dex-P.

The images corresponding to DMPC monolayers showed nucleation of condensed domains (light gray spots) at about 33 mN/m, in agreement with the phase transition observed at this surface pressure (Figure 2a). At 35 mN/m, a homogeneous condensed phase was observed (Appendix A).

Figure 4a,b shows the Rp values as a function of the surface pressure. The Rp value increased with surface pressure as a consequence of the change in the hydrocarbon tail orientation and film density, and at 33 mN/m, Rp increases abruptly due to phase transition going from a mean value of (0.67 ± 0.04) × 10^−6^ for the LE phase to (1.5 ± 0.07) × 10^−6^ for the LC phase, Figure 4a.

Dex or Dex-P incorporation on DMPC monolayers had several consequences that can be seen in the BAM images. Increasing the drug molar fraction modified the LC phase nucleation process, affected the reflectivity values, and induced the formation of aggregates. Each of these observations are detailed in what follows.

#### 3.3.1. Modification of the LE/LC Phase Transition

The effect promoted by the drug on the film phase transition depended on the molecule. As already discussed, Dex-P did not erase the phase transition of DMPC monolayers, and thus, the nucleation process was not affected (Figure 3b and Figure 4a).

However, when Dex-P was present, the range of pressure at which the phases coexist was an increase of 1.5 mN/m, ranging from 33–35 mN/m in absence of Dex-P to 31.5–35 mN/m for a Dex-P molar fraction of 0.8 (Appendix A).

A similar effect over nucleation was found for Dex until a 0.5 molar fraction, where the pressure range of LE/LC phase coexistence was 32–36 mN/m, 2 mN/m higher that pure DMPC, Appendix A. A further increment of Dex amounts (DMPC: Dex 3:7) completely avoided the LC domain nucleation (Figure 3a) in agreement with the decrease in compressibility at high surface pressure (Figure 2c,f) and the disappearance of the LE/LC transition (Figure 2a). Dex presence between DMPC hydrocarbon tails avoided the hydrophobic interaction between them, inhibiting the passage of LE to LC phase. An expansion in DPPC monolayer cause by Dex was previously reported for DPPC monolayers and attributed to the presence of 3-Keto group that destabilize the membrane model [41], Figure 1a. Besides, Wenz et al. found that corticosteroids without the aliphatic chain in 17-C, Figure 1a, have a lipid domain disrupting activity [42].

Regarding the increment of the LE/LC coexistence region in DMPC: Dex or Dex-P mixtures, this is an usual behavior of mixtures [43], in which the presence of a new component increments the range of the variable parameter at which two phases coexist compared to the pure component.

#### 3.3.2. Changes in the Reflectivity of Films in LE and LC Phases

Figure 4a,b shows the reflectivity obtained from the BAM images as a function of the lateral surface pressure for DMPC: Dex and DMPC: Dex-P monolayers at all the studied molar fractions.

As can be seen in Figure 4a, the presence of Dex-P in the films did not modify the Rp values in the whole range of surface pressures studied compared to pure DMPC monolayers, independently of the molar fraction analyzed.

Contrarily, Dex presence induced a decrease in Rp values for molar fractions higher than 0.1 Dex and for pressures higher than 35 mN/m (LC phase for Dex molar fraction ≤0.5), Figure 4b,c. As demonstrated in Figure 4c, no Rp modification was found in a surface pressure range of 20–25 mN/m, while a concomitant reduction in Rp value with the increase in Dex fraction in the range of 40–45 mN/m was evident. These results agree with the decrease in compressibility at high surface pressure, the less narrower phase transition and the loss of the LC phase nucleation process after 0.5 Dex fraction. These effects can be explained considering that the Dex presence between DMPC hydrocarbon tails avoid the hydrophobic interaction between them inhibiting the transition from LE to LC phase, causing a reduction in film compressibility and reflectivity.

#### 3.3.3. Formation of Aggregates at the Interface

In addition to the changes discussed above, BAM images obtained for DMPC:X (X: Dex or Dex-P molar fraction ≥0.1) films showed the presence of very bright dots (Figure 3a,b, orange arrows) dispersed non homogeneously on the monolayer. The high reflectivity of these dots suggest that they correspond to 3-dimensional aggregates (structures thicker than a monomolecular film), which are in agreement with the reduction in the area per spreaded molecule shown in Figure 2e. The aggregates appeared at low surface pressures and had characterized for a mean size between 6–12 μm^2^ and 5–7 μm^2^ for Dex-P and Dex, respectively, that did not change with surface pressure (result not shown) or with the molar fraction of the drug (Appendix A). It can be seen that the aggregates generated by the presence of Dex-P had slightly larger areas than those generated in the presence of Dex, for molar fractions >0.1 (see Appendix A).

Figure 5a,b shows the average number of aggregates quantified at three ranges of surface pressures at different drug molar fractions. An increase in the number of aggre-gates was observed when the molar fraction of the drug and the surface pressure in-creased. In the case of Dex, there was an abrupt change in the number of aggregates at a molar fraction of 0.7, while for Dex-P, it increased more gradually. This difference could be because Dex partially mixed with DMPC up to a molar fraction of 0.8 where it segregated. Dex-P mixed with DMPC in a lower degree, since it did not induce modifications on film properties. The number of aggregates was slightly higher for Dex-P than for Dex up to a fraction of 0.8.

Subsequently, the Rp value of the aggregates was analyzed for each monolayer as a function of surface pressure at fixed drug molar fraction. We found that there was not-linear relationship between Rp and surface pressure either for Dex or Dex-P at all molar fraction (see Appendix A). Therefore, the Rp value of the aggregates was not modified during compression. Appendix A shows the frequency histogram for Rp at the different drug molar fractions, for Dex and Dex-P, for all surface pressures. As can be seen from this figure, there was a main population with a maximum reflectivity, followed by less frequent Rp values. The most probable distribution was adjusted with a gaussian function to obtain the most probable Rp value for each condition, and these values are summarized on Figure 5c.

If we compare the Rp values of the aggregates with those of the condensed phase of DMPC, we can estimate the relative thickness between the spots and the monolayer assuming that the refractive index of the monolayer of DMPC and the spots are similar. The square root of the ratio between reflectivity ($\sqrt{\mathrm{Rp}\;\mathrm{aggregates}/\mathrm{Rp}\;\mathrm{DMPC}}$) corresponds to the ratio in film thicknesses [44]. These values were larger than 1 but lower than 2, which means that the aggregates did not correspond to segregation of DMPC in a condensed monolayer ($\sqrt{\mathrm{Rp}\;\mathrm{aggregates}/\mathrm{Rp}\;\mathrm{DMPC}}$) = 1) or bilayer ($\sqrt{\mathrm{Rp}\;\mathrm{aggregates}/\mathrm{Rp}\;\mathrm{DMPC}}$) = 2). With this in mind, we propose that the aggregates correspond to segregated Dex or Dex-P molecules, with different refractive indexes and thus, different Rp values [45]. Assuming that the refractive index of the aggregates was similar to that reported for cholesterol or ergosterol (between 1.443 and 1.448, [45]), we obtained an aggregate thickness of 3 nm, which corresponds to Dex or Dex-P multilayers (Dex length in a crystal is 1.1 nm, [46]). This is the case for the most probable Rp values, the less probable Rp values would correspond to thicker drug structures that occurred with less probability.

There is evidence that Dex-P can aggregate in solution by π-π stacking due to the interaction between the π system present in its “A” hexene ring [47], Figure 1. Additionally, progesterone (structural similar steroid) can form co-crystal in solid state by the formation of α − π stacking [48]. Similar aggregates were observed for pure cholesterol when the monolayer is near to the collapse pressure [37,49] or in a mixed monolayer of DPPE: Chol when the molar percentage of cholesterol is higher than 60%. Under this last condition, the mixture DPPE and Chol do not mix [50]. Bhardwaj et al. found that Dex and Chol distribute heterogeneously in PC liposomes at high concentrations [4]. With all this evidence, and considering that all corticoids derive from cortisol, which in turn is synthetized from cholesterol (this molecule has the same steroid core than cholesterol), we propose that the domains present at high drug concentrations correspond to three-dimension aggregates enriched in Dex or Dex-P.

### 3.4. Z-Potential and Hydrodynamics Measurements

To corroborate the adsorption of Dex-P on bilayers, LUVs with an average diameter of 100 nm were exposed to increasing Dex-P concentrations, and the potential at the slipping plane (ζ, z-potential) was determined after 1 h of exposition to the drug. All the measurements were done in NaCl 1 mM in order to maximize the Debye length, so that z-potential can be considered similar to the surface potential of the vesicle.

Figure 6 shows the ζ value and the hydrodynamic size as function of Dex-P/lipid molar fraction. DMPC vesicles in the absence of the drug had a negative value of ζ, (−40 ± 4) mV, which coincide in sign with other published reports for PC vesicles in NaCl 1 mM [51,52,53]. When Dex-P was added to the DMPC LUVs suspensions at Dex-P/lipid = 6 ratio, the ζ value decreased to a value of (−55 ± 3) mV (a final drug concentration of 0.72 mM). Further additions of the drug did not alter ζ. This indicates that Dex-P adsorbed on DMPC vesicles, leading to more negative surface potentials due to the negative charge of the drug and that saturation is reached at 0.72 mM of Dex-P (Dex-P/lipid = 6). This result confirms the adsorption of Dex-P on DMPC bilayer. The Dex-P adsorption does not modify the hydrodynamic size of DMPC vesicles.

### 3.5. Giant Unilamellar Vesicles (GUVs)

With the aim of understanding the action of Dex and Dex-P adsorption on membrane mechanical properties, more specifically membrane rigidity, we studied the shape fluctuation of individual GUVs before and after the addition of the drugs registering vesicle shape with confocal fluorescence microscopy.

The bending rigidity constant of a bilayer is particularly interesting because it is the energy needed to change the membrane curvature of a bilayer and can be altered by changes in the thickness of the membrane [31,54,55,56], the thickness of the double layer [56] and on the surface charge of the membrane [57,58,59]. Briefly, an increment in the membrane thickness, or in the surface charge or in the double layer distance increments the energy necessary to bend the bilayer, and the larger is the value of the bending rigidity, the stiffer is the membrane.

With the purpose of qualitatively analyzing the effects of the drugs on bending rigidity, we determined the shape fluctuation of individual GUVs at two different conditions: (1) GUVs in NaCl 10 mM with and without 0.2 mM of Dex, in this experiment the drug was added from an ethanol solution, therefore the control vesicles (without Dex) were performed with the same volume of ethanol (5 µL to 145 µL, 3% *v*/*v*); and (2) GUVs in NaCl 10 mM with and without 1 mM of Dex-P. Each individual vesicle was recorded during 500 frames (460 frames/min) and the $\bar{Cir}$ and $\mathrm{SD}\bar{Cir}$ respect to time were obtained, Appendix A. After this, the vesicles were classified in fluctuating or non-fluctuating as described in the experimental Section 2.2.2. Figure 7a shows a representative image for a fluctuating or a non-fluctuating vesicle as a function of the time.

Figure 7b, shows the percentage of fluctuating GUVs in the absence and presence of 0.2 mM Dex. In the absence of the drug, the percentage of fluctuating vesicles was of (7 ± 2) %, and in the presence of Dex was of (10 ± 3) %. In this case, there was no statistically significant difference between both average values. Previous reports of the action of ethanol in DMPC or DPPC vesicles showed a shape change in the vesicle (due to an interdigitation of the hydrocarbon chains), but these changes were found at 5% *v*/*v* [60,61]. In this work, the addition of ethanol was 3% *v*/*v* and no vesicle shape deformation was found, but it cannot be discarded that ethanol intercalates in the hydrocarbon tails and modifies drug adsorption. Therefore, we cannot rule out that Dex did not modify the membrane rigidity since the presence of ethanol may hide the effects of Dex.

Contrary to what it has been found with Dex, the resuspension of the GUVs in a solution containing 1 mM of Dex-P reduced the percentage of fluctuating vesicles from (36 ± 2) % in absence to an (8 ± 3) % in the presence of Dex-P, Figure 7b. A decrease in the population of fluctuating vesicles, suggests a more rigid membrane, and, consequently, a lager bending rigidity constant. This result can be explained taking into account the results show previously: Dex-P adsorbed in DMPC monolayer (Section 3.1) and also in DMPC LUVs (Section 3.4, Figure 6) making more negative the z-potential value, indicating that Dex-P charged negatively the bilayer. Dex-P did not induce an evident change in monolayer thickness (no modification in the reflectivity of DMPC films, Section 3.3) or liposome size, Figure 6 Section 3.4. Therefore, the presence of Dex-P only increased the surface charge and the electrostatic repulsion, making the membrane bending process energetically more expensive and inducing a stiffening of the membrane.

## 4. Discussion

Regarding the insertion experiments, we found that both drugs adsorb to DMPC monolayers, and Dex had a higher affinity for the lipid monolayer than Dex-P due to its higher hydrophobic character, but both drugs can be inserted in the monolayers, even at high surface pressures. This fact indicates that the drugs can insert spontaneously in a condition of similar lipid density of a biological membrane. In this context, Dex induced larger changes in membrane packing than Dex-P.

Taking into account the insertion experiments, Figure 1b and the Langmuir isotherm vs. the area per lipid (Appendix A), we estimated the adsorption percentage of the drug at a defined initial pressure. At an initial surface pressure of 10 mN/m, the final change in pressure was 16 mN/m and 5 mN/m for Dex and Dex-P, respectively (Figure 1b). Considering this result and the Langmuir compression isotherms (Appendix A), the corresponding adsorption fraction of the drug corresponds to a molar fraction of 0.7 for Dex and 0.1 for Dex-P (these are the drug molar fractions corresponding to each change in surface pressure). Dex-P adsorbed spontaneously in a lesser proportion than Dex, an effect that could be due not only to a lower affinity for the hydrophobic core of DMPC films, but also to the presence of the negative phosphate group since, once Dex-P adsorbs to the DMPC monolayer, the film becomes negative charged, hindering the adsorption of additional Dex-P molecules due to electrostatic repulsion.

From Langmuir isotherms and Brewster angle microscopy images, we found that the changes in the monolayer properties depend on the nature of the studied drug. The presence of Dex mixed with DMPC makes the film more compressible at surface pressures higher than 35 mN/m, reducing the Rp value, making the LE/LC phase transition less marked, and also increasing the range pressure of LE/LC phase coexistence until the phase transition is no longer evident (at Dex molar fraction >0.5). BAM images evidenced the presence of aggregates at all molar fractions of Dex, increasing in number with the increment of Dex fraction in the film. The appearance of aggregates is in concordance with the reduction in area per spreaded molecules of the Langmuir isotherms as Dex amount increased. From images quantification, we found that aggregates have higher Rp reflectivity values than DMPC film. With this evidence, we postulate that aggregates correspond to Dex segregation in 3-dimensional structures, that are more evident after 0.7 Dex molar fraction. Figure 2 summarizes our hypothesis. A fraction of Dex initially partially mixed with DMPC molecules modifying the monolayer properties aforementioned and disrupting the LC phase formation of DMPC. At molar fractions higher than 0.7, Dex abruptly segregates increasing the number of aggregates dramatically, probably due to an immiscible mixture between Dex and DMPC.

Regarding Dex-P, its presence slightly increases the compressibility modulus of the condensed phase, but does not affect the LE/LC phase transition of DMPC nor the reflectivity of the film at all surface pressures. Dex-P aggregates at all molar fractions, not affecting the monolayer properties. Probably the fraction partially mixed with DMPC monolayer prolapse to the water subphase, not affecting DMPC molecules in the LC phase.

Finally, from z-potential experiments, we corroborated that Dex-P adsorbs onto DMPC vesicles, making them more negatively charged. From the analysis of membrane fluctuation in DMPC GUVs, we found that the adsorption of Dex-P induces a decrease in the population of fluctuation vesicles due to the presence of the negative charge in its surface, which makes the bending of the membrane more energetically expensive, stiffening the vesicle membrane.

## 5. Conclusions

Given the importance that Dex and Dex-P has acquired in recent years, it is relevant to deepen the understanding of the effects and counter effects of their action on cellular membranes. Its anti-inflammatory and immunosuppressive effect has been widely characterized, but less is known about its action on cell membranes, which is highly important, since it is the first barrier for their action.

This work represents a first approach for understanding the interaction of Dex and Dex-P with lipids. Using simple systems, we described how this drug modifies the mechanical properties of membranes. We found that both drugs form 3-dimentional aggregates in DMPC: drug monolayers. Dex modifies the compressibility moduli of DMPC monolayers, which may also occur in other films such as pulmonary surfactant. This would affect the compression-expansion processes that occur in lungs during breathing, thus compromising alveoli collapse. Since Dex-P affects lipid membrane out-of-plane fluctuations, this drug may affect the stiffness of the lipidic region of the cells for high local drug dose.

Our finding that Dex and Dex-P form 3-dimensional aggregates at molar fractions of 0.1 or higher suggests that in chronic treatments, when the drugs are administered during long periods, they may accumulate in cellular membranes. Therefore, the results shown here may contribute to understanding side effects of the drugs related to modifications of the lipid bilayer of membranes.

## Data Availability

Not applicable.

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
