# Peer review of "Dexamethasone and Dexamethasone Phosphate: Effect on DMPC Membrane Models"

_pharmaceutics, 2023, doi:10.3390/pharmaceutics15030844_

Round 1
Reviewer 1 Report
The study deals with the adsorption of Dexamethasone (Dex) and Dexamethasone phosphate (Dex-P) on dimyristoylphophatidylcholine (DMPC) monolayers and aims to understand how the presence of the drugs modifies the interfacial film properties.
Specific comments
- The authors provide convincing evidence that both drugs can penetrate and alter the interfacial properties of DMPC but the physiological relevance (extensively discussed in the manuscript) of the study is highly questionable. Indeed the alveolar monolayer is mostly enriched in DPPC (to a large extent 80 mol% or greater) and DMPC is not at all representative of the lipid composition of pulmonary alveolar sacs with which Dex and/or Dex-P will first interact.
- All experiments are carried below the DMPC transition temperature (23C°) and both drugs penetrate a rigid lipid film. Experiments carried out above the transition temperature will lead to the compressional behavior of Langmuir fluid monolayers.
- Phosphatidylglycerol (PG) is an unsaturated anionic component of the lipid surfactant composition and will drastically limit DeX-P-lipid interactions. Again the lipid composition of the model membrane chosen by the authors (DMPC) does not give access to any physiologically relevant information.
Author Response
Review Comments
Reviewer #1:
The study deals with the adsorption of Dexamethasone (Dex) and Dexamethasone phosphate (Dex-P) on dimyristoylphophatidylcholine (DMPC) monolayers and aims to understand how the presence of the drugs modifies the interfacial film properties.
Specific comments
R./ The authors thank the reviewer for the revision of the paper, and for their suggestions to improve the article.
- The authors provide convincing evidence that both drugs can penetrate and alter the interfacial properties of DMPC but the physiological relevance (extensively discussed in the manuscript) of the study is highly questionable. Indeed the alveolar monolayer is mostly enriched in DPPC (to a large extent 80 mol% or greater) and DMPC is not at all representative of the lipid composition of pulmonary alveolar sacs with which Dex and/or Dex-P will first interact.
We agree with the referee's comment and have rewritten the introduction section. The performed experiments are far from describing pulmonary surfactant-drug interaction, however they give an insight of the physicochemical characteristics of lipid-drug interaction in a simple system.
- All experiments are carried below the DMPC transition temperature (23C°) and both drugs penetrate a rigid lipid film. Experiments carried out above the transition temperature will lead to the compressional behavior of Langmuir fluid monolayers.
GUVs and LUVs experiments were performed at 25 C (the manuscript had a mistake, we corrected this).
Only the monolayer experiments were performed at temperatures below 23 C, in order to inquire about the effect of the drugs on the phase transition. Monolayers are fluid up to the surface pressure corresponding to the phase transition (30-35 mN/m at 15 C), therefore in these experiments we are able to study drug penetration both, in fluid and rigid films.
Insertion experiments were performed at 20C, at this temperature monolayers are fluid up to 42 mN/m.
- Phosphatidylglycerol (PG) is an unsaturated anionic component of the lipid surfactant composition and will drastically limit DeX-P-lipid interactions. Again the lipid composition of the model membrane chosen by the authors (DMPC) does not give access to any physiologically relevant information.
We agree with the referee's comment in that a more complex lipid composition would be more comparable to pulmonary surfactant. However, results obtained with complex systems are more difficult to interpret and thus, we started with a simple system.

Reviewer 2 Report
This paper discusses the effect of dexamethasone and dexamethasone phosphate on DMPC model lipid bilayers and monolayers. Some comments and suggestion for the authors:
(1) English language needs to be improved. The professional proofreading is highly recommended
(2) This paper has a significant number of typos (e.g line 133 1nmol.mL-1 should be
1nm×mL-1 or 1nm/mL; line 331: mN.m-1 should be mN×m-1 or mN/m, figure 2. area per sedeed should be area per seeded)
(3) Figure 1 (b). It is necessary to explain in the figure caption what do the arrows on the graph mean.
(4) Readers have no idea what is the molecular area per lipid and the area per seeded molecule. Explanation suggested.
(5) Figure 7(b) is too small.
(6) Figure caption “DMPC: Dex-P 0.2:0.8 LE/LC phase transition and aggregates” change to DMPC: Dex-P 0.2:0.8 LE/LC phase transition and Dex-P aggregates and Dex-P monomers.
(7) Conclusion: Cell membrane show complex behavior. This paper do not “show
a first approach to understand the action of Dex and Dex-P on cellular membranes”. It should also be pointed out that experiments have been carried out at temperature below the main phase transition. The fraction of aggregates depends on the physical state of lipid phase and should be lower for fluid phase.
Summary recommendation. This manuscript could be published after the revisions suggested above.
Author Response
Review Comments
Reviewer #2:
This paper discusses the effect of dexamethasone and dexamethasone phosphate on DMPC model lipid bilayers and monolayers. Some comments and suggestion for the authors:
R./ The authors thank the reviewer for the revision of the paper, and for their suggestions to improve the article.
(1) English language needs to be improved. The professional proofreading is highly recommended
We have revised the whole manuscript and the English language has been improved.
(2) This paper has a significant number of typos (e.g line 133 1nmol.mL-1 should be 1nm×mL-1 or 1nm/mL; line 331: mN.m-1 should be mN×m-1 or mN/m, figure 2. area per sedeed should be area per seeded)
The manuscript was revised thoughtfully, and these errors have been corrected in the main text and in the Figures.
(3) Figure 1 (b). It is necessary to explain in the figure caption what do the arrows on the graph mean.
Done. The sentence ¨ Arrows indicate the πcut off values for each drug¨ was added, line 254-255.
(4) Readers have no idea what is the molecular area per lipid and the area per seeded molecule. Explanation suggested.
Sorry for this mistake, the more often term is spreaded. We now changed this word and explained the term in the experimental part (line 137-139).
(5) Figure 7(b) is too small.
The figure has been modified.
(6) Figure caption “DMPC: Dex-P 0.2:0.8 LE/LC phase transition and aggregates” change to DMPC: Dex-P 0.2:0.8 LE/LC phase transition and Dex-P aggregates and Dex-P monomers.
Caption was changed to ¨ Schematic representation of (a) Pure DMPC in the LE and the LC phase. (b) DMPC: Dex 0.2:0.8, Dex mixed with DMPC in the LE phase and Dex aggregates. (c) DMPC: Dex-P 0.2:0.8, Dex-P mixed with DMPC in the LE phase, Dex-P aggregates and DMPC in the LC phase ¨.
(7) Conclusion: Cell membrane show complex behavior. This paper do not “show a first approach to understand the action of Dex and Dex-P on cellular membranes”. It should also be pointed out that experiments have been carried out at temperature below the main phase transition. The fraction of aggregates depends on the physical state of lipid phase and should be lower for fluid phase.
Conclusion section was rewritten considering the referee's comment. Aggregates form in monolayers at surface pressure below that corresponding to phase transition. Films are in a fluid state.
Summary recommendation. This manuscript could be published after the revisions suggested above.

Round 2
Reviewer 1 Report
.